# Chemometric Models of Differential Amino Acids at the Na_v_α and Na_v_β Interface of Mammalian Sodium Channel Isoforms

**DOI:** 10.3390/molecules25153551

**Published:** 2020-08-03

**Authors:** Fernando Villa-Diaz, Susana Lopez-Nunez, Jordan E. Ruiz-Castelan, Eduardo Marcos Salinas-Stefanon, Thomas Scior

**Affiliations:** 1Laboratory of Computational Molecular Simulations, Faculty of Chemical Sciences, BUAP, C.P. 72570 Puebla, Mexico; fer.vdl1928@gmail.com (F.V.-D.); or hefzi-ba@live.com.mx (S.L.-N.); toj_cdai90@live.jp (J.E.R.-C.); 2Institute of Physiology, BUAP, C.P. 72570 Puebla, Mexico; eduardo.salinas@correo.buap.mx

**Keywords:** homology modeling, extracellular loops, interactome, protein-protein interaction, hot spot prediction

## Abstract

(1) Background: voltage-gated sodium channels (Na_v_s) are integral membrane proteins that allow the sodium ion flux into the excitable cells and initiate the action potential. They comprise an α (Na_v_α) subunit that forms the channel pore and are coupled to one or more auxiliary β (Na_v_β) subunits that modulate the gating to a variable extent. (2) Methods: after performing homology in silico modeling for all nine isoforms (Na_v_1.1α to Na_v_1.9α), the Na_v_α and Na_v_β protein-protein interaction (PPI) was analyzed chemometrically based on the primary and secondary structures as well as topological or spatial mapping. (3) Results: our findings reveal a unique isoform-specific correspondence between certain segments of the extracellular loops of the Na_v_α subunits. Precisely, loop S5 in domain I forms part of the PPI and assists Na_v_β1 or Na_v_β3 on all nine mammalian isoforms. The implied molecular movements resemble macroscopic springs, all of which explains published voltage sensor effects on sodium channel fast inactivation in gating. (4) Conclusions: currently, the specific functions exerted by the Na_v_β1 or Na_v_β3 subunits on the modulation of Na_v_α gating remain unknown. Our work determined functional interaction in the extracellular domains on theoretical grounds and we propose a schematic model of the gating mechanism of fast channel sodium current inactivation by educated guessing.

## 1. Introduction 

### 1.1. The Sodium Channels

Concerning the voltage-gated sodium channels (Na_v_s), a plethora of genes, their reading frames, expression patterns and functions have been reported for various organisms ranging from prokaryotic to eukaryotic cells. We treat the isoforms of ion channels in vertebrates with their greater gene complexities [1,2].

The Na_v_ complex generally consists of a central α (Na_v_α) subunit with the channel pore that is encoded by *SCN1A* to *SCN5A* (Na_v_1.1α to Na_v_1.5α, respectively) and *SCN8A* to *SCN11A* (Na_v_1.6α to Na_v_1.9α, respectively). The nine isoforms of the Na_v_α subunit are expressed in specific tissue patterns and exhibit differences in gating behavior that adapts them to different physiological functions [3,4,5,6]. 

The pharmacological classification of these subtypes diverges according to their sensitivity and resistance to tetrodotoxin (TTX); Na_v_1.1α to Na_v_1.4α, Na_v_1.6α and Na_v_1.7α are sensitive to blocking by low nanomolar concentrations of TTX (TTX-S) and Na_v_1.5α, Na_v_1.8α and Na_v_1.9α are resistant to concentrations >1 μM TTX (TTX-R) [7].

More than a thousand point mutations have been identified in human Na_v_s, while some of them have been associated with neurological, cardiovascular, muscular, and psychiatric disorders, such as epilepsy, arrhythmia, muscular paralysis, pain syndrome, and a broad spectrum in autism disorder [8,9,10,11,12]. 

Na_v_s are targets for a wide variety of natural toxins and clinical therapeutic drugs [13,14,15,16]. 

### 1.2. The Na_v_α Subunit 

The Na_v_α subunit of vertebrates consists of a single polypeptide chain with an approximate molecular mass of 260 kDa. It embraces the ion selective component that folds into four homologous but not identical domains (DI to DIV), each domain contains six transmembrane helical segments (S1 to S6), which are assembled around the ion selective pore [17,18]. 

The transmembrane helical segments S1, S2, S3, and S4 comprise the four voltage sensing domains (VSDs) in DI to DIV. They are located at the outer edge at each corner of the Na_v_α subunit (Figure 1). The S4 helix constitutes the voltage sensor of each VSD. It has evolved into an amphipathic domain with a positively charged face. In response to the changes in the electric field produced by the depolarization of the membrane, the S4 moves towards the extracellular zone initiating conformational changes, which in turn open the pore [19,20,21,22]. The three S4 of the DI, DII, and DIII show faster kinetics in response to depolarization and allow sodium cations to enter the cells. The S4 of the DIV responds more slowly. Its movement releases an intracellular connector called the IFM inactivation gate. It contains three lipophilic residues (isoleucine, phenylalanine, methionine), and connects the S6 DIII to S1 DIV [23,24,25]. As a result, the inactivation gate moves to occlude the pore and leads the channel into an inactive state. Therefore, the activation and inactivation of the channel are linked in a structural, mechanical, and functional way [26,27].

### 1.3. The Na_v_β Subunits

Most Na_v_s of vertebrate cells form biological units with associated β subunits (Na_v_βs). There are four Na_v_β genes (*SCN1B* to *SCN4B*) that encode four proteins Na_v_β1 to Na_v_β4, respectively [28,29]. Like the Na_v_α subunits, the Na_v_β subunits are individually expressed for tissue differentiation [30,31].

While the Na_v_α subunit is sufficient for voltage detection and selective ion conductance, the Na_v_βs subunits modulate peak values of sodium current and modify the kinetics of the activation and inactivation of the Na_v_α subunit. They increase peak current density by augmentation of the channel density (number per area) on the cell surface. They effectively change the voltage range involved in activation and inactivation and improve inactivation and recovery rates of inactivation [5,28,32,33,34,35,36,37,38,39].

Prior to this chemometric study we carried out electrophysiological and site-directed mutagenesis experiments combined with molecular modelling to study modulation of Na_v_α subunit by Na_v_β1 [40,41,42].

All Na_v_β subunits are type 1 membrane proteins. The extra-cellular amino-terminal region contains a single V-type amino-terminal immunoglobulin domain (IgD) and a short neck connected to a transmembrane helix (TMH) in addition to a carboxy-terminal intracellular region. The sequences similarities between Na_v_β1 and Na_v_β3 are higher than between Na_v_β2 and Na_v_β4 [43,44]. Na_v_β1 and Na_v_β3 are linked to Na_v_α through non-covalent interactions, while Na_v_β2 and Na_v_β4 are linked by a disulfide bridge with Na_v_α [45,46].

The cryo-electron microscope (cryo-EM) structures of Na_v_s of insects, electric eel, rat, or human species reveal an identical three-dimensional (3D) architecture of their Na_v_α subunits. Comparing the interface between Na_v_α and Na_v_β (Na_v_α/Na_v_β) of electric eel as well as *Homo sapiens* reveals that it is astonishingly well conserved [47,48,49,50,51,52].

The identification of the interaction sites for modulation of the Na_v_α subunit with the Na_v_β subunits mainly stems from the mutagenic analysis and structural information of the individual Na_v_β subunits [36,40,41,42,46,49,53,54,55,56,57,58].

### 1.4. The Present Contributions

Currently, it remains an unanswered question how exactly the pore subunit (Na_v_α) is modulated by the Na_v_β1 and Na_v_β3 subunits. Our present chemometric study aims at describing mechanistic behavior at the Na_v_α/Na_v_β to shed light on the modulation by the nine isoforms (Na_v_1.1α to Na_v_1.9α) comparing three Mammalian species, namely *Homo sapiens* (hNa_v_), *Mus musculus* (mNa_v_) and *Rattus norvegicus* (rNa_v_). To this end, well-established in silico methods were applied, like multiple sequence alignments, structural determinations, amino acid properties analysis, the generation of homology models as well as molecular electrostatic potentials, which can be color coded and projected on the molecular surfaces (MEPS). 

To suit our chemometric analysis, the pore-bearing α subunit is dissected in the following topological parts. Herein after the entire pore subunit will be denominated as Na_v_α or α for short. It is in turn composed of three topological segments: (i) the extracellular part or region (ECR) with its 16 extracellular loops (ECLs); (ii) the transmembrane helical part (TMH); and finally (iii) the intracellular region. 

On theoretical ground, we determined (3D) structural features and sequence patterns as well as atom properties to describe the protein-protein interactions (PPI) between two pairs of proteins: not only Na_v_α with Na_v_β1 subunits (Na_v_α/Na_v_β1) but also Na_v_α with Na_v_β3 subunits (Na_v_α/Na_v_β3). The term PPI implies that both pairs were always treated in parallel for all three Mammalian species to provide a total and systematic view on chemometric patterns. While the (3D) structures of the former pair have been experimentally elucidated (by cryo-electron microscopy or crystallography), no (3D) structural information exists for the latter pair. This means, on the one hand we studied existing Na_v_α/Na_v_β1 complexes, while on the other hand we directly applied our findings to create a hitherto unknown interface (symbol /) between Na_v_α with Na_v_β3 (Na_v_α/Na_v_β3). Herein after, the observed as well as the postulated interface will be denominated as IF, for short. The ectodomain of said IF contains the extracellular loops of the Na_v_α subunit, which will be called ECLs in our study. All those (short) ECL segments in interaction with Na_v_β1 or Na_v_β3 subunits will be designated as IF-ECLs. All told, ECLs always belong to an α subunit, never to a β subunit because the latter does not possess loops, only antiparallel beta-strands which bend in turns and hair pins to form ordered beta-sheets (immunoglobulin domain, IgD, or all-beta fold). Finally, IF-ECLs embrace short amino acid segments, sometimes only a few individual residues, which we studied at either a sequential or even atomic scale. 

## 2. Results 

### 2.1. Determination of PPI in the Isoforms of the Na_v_s 

Our chemometric study produced detailed data, a fairly larger portion of which we present in the Appendix A section. Precisely, the observed as well as the computed property patterns of interacting residues at the interface were described for eight PPI patches (thereupon called PPI-Id) by nine isoforms of three species for two pairs (Na_v_α/Na_v_β1 and Na_v_α/Na_v_β3), all of which yields 832 PPIs in 27 3D models (8 × 9 × 3 × 2) based on known 3D structures. When we take into account the four domains on each α subunit, the pore chain and the fact that each of those four domains (D1 to DIV) exposes four ECLs, then the sheer amount of 432 sets of calculations (16 loops × 9 isoforms × 3 species) were prepared, carried out, gathered, documented and interpreted. Table 1 provides a synopsis about the obtained results describing the interaction between Na_v_α/Na_v_β1 or Na_v_α/Na_v_β3. Of note, eight different computed polar interaction patterns were identified across all four domains (DI to DIV). They extend by far the extant literature for its systematic analysis and completeness (cf. 2.1.1.).

In Table 1 nine PPI patterns (Roman numerals) were detected as a result of the line-wise combination of Yes/No interaction features. Each line represents the eight identified potential contact zones in our study (PPI-Ids): (**I**) Id 6 has no PPI for hNa_v_, mNa_v_ and rNa_v_ (1.1, 1.3 and 1.7); (**II**) Ids 1 to 8 have PPIs for hNa_v_, mNa_v_ and rNa_v_ (1.2, 1.4 and 1.6); (**III**) Id 3 and 6 have no PPIs for hNa_v_, mNa_v_ and rNa_v_ (1.5); (**IV**) Id 4 and 6 have no PPIs for hNa_v_1.8α; (**V**) Id 3, 4 and 6 have no PPI for mNa_v_1.8α; (**VI**) Id 2, 3, 4 and 6 have no PPI for rNa_v_1.8α; (**VII**) Id 2, 4, 6, 7 and 8 have no PPI for hNa_v_1.9α; (**VIII**) Id 3, 4, 6, 7 and 8 have no PPI for m, rNa_v_1.9α; (**IX**) Id 1 has no PPI for eeNa_v_1.4α. 

#### 2.1.1. PPI Analysis in the Structural Complex eeNa_v_1.4α/eeNa_v_β1 

Figure 2 presents the PPI for eeNa_v_1.4α/eeNa_v_β1 (PDB: 5XSY [48]). Six interactions in three ECLs had already been published prior to our study: S1–S2 DIII, S5 DIV, and S6 DIV. However, our analysis of eel template unveiled a hitherto unpublished interaction site on ECL S5 DI (6 + 1 = 7 PPI-Ids).

#### 2.1.2. PPI Analysis of the hNa_v_1.4α/hNa_v_β1 and hNa_v_1.4α/hNa_v_β3 Models 

The 3D template complexes eeNa_v_1.4α/eeNa_v_β1 [48] and hNa_v_1.4α/hNa_v_β1 [49] possess a significant sequence identity (Na_v_α ≈ 65% and Na_v_β ≈ 46%, resp.) in addition to a relatively high degree of conserved residues by homology. A measure of geometrical deformation is the so-called root-mean-square deviation (RMSD). The PDB entries 5XSY [48] versus 6AGF [49] were compared in terms of RMSD for both subunits: Na_v_α ≈ 0.942, Na_v_β ≈ 0.955. A first inspection of both 3D templates by eyesight also revealed how well-conserved are all IF-ECLs with Na_v_β1 between both species. 

After the detection of the seventh PPI site (cf. 2.1.1.) we inspected the 3D template hNa_v_1.4α/hNa_v_β1 [49] and our homology model hNa_v_1.4α/hNa_v_β3 [49,55] (Figure 3). At this stage we detected another PPI site on the ECL S5 DI. Hereupon we label the six published and the two detected PPI sites as follows: PPI-Ids 3 to 8 and PPI-Ids 1 and 2, respectively. Of note, PPI-Ids 3 to 8 were observed on 3D template eeNa_v_α1.4/eeNa_v_β1 [48]. Albeit, side chain rotation by Chimera’s built-in rotamer library [59,60,61] had to be applied on the 3D template hNa_v_1.4α/hNa_v_β1 [49]. Both detected PPI-Ids (1 and 2) lie on ECL S5 DI.

#### 2.1.3. Identification of the Interacting Residues

Multiple sequence alignment (MSA) was performed by the Web-based Clustal Omega server [62] with the primary sequences of eeNa_v_1.4α, hNa_v_1.1α to hNa_v_1.9α, mNa_v_1.1α to mNa_v_1.9α, rNa_v_1.1α to rNa_v_1.9α and all β subunits (eeNa_v_β1, hNa_v_β1 to hNa_v_β4, mNa_v_β1 to mNa_v_β4 and rNa_v_β1 to rNa_v_β4). We identified the conserved amino acids and replacements by homology since they constitute either structurally or functionally pivotal components for the channel (Table 2). As a most valuable asset, the hitherto known interacting residues of the 3D templates served as references to identify the interacting residues (Table 1 and Appendix A). The 3D templates were as follows:eeNa_v_1.4α/eeNa_v_β1, hNa_v_1.4α/hNa_v_β1, and hNa_v_1.4α/hNa_v_β3 (Figure 2 and Figure 3). 

On Na_v_α two unchanged residues were detected in sequence positions labeled as PPI-Ids 1 and 5 on ECLs S5 DI and S5 DIV, respectively. On the counter subunit β, both subunits contain partially conserved residues in positions PPI-Ids 2b, 3b, and 5b, but on PPI-Id 8b valine (V) remains unchanged. Intriguingly both βs also present two homologous residues by keeping their respective charges at positions 4 (negative) and 6 (positive) in all isoforms and species. In contrast, Table 2 also unravels that equivalent residues on Na_v_β2 or Na_v_β4 are not conserved when compared to locations on the three reference sequences of the eeNa_v_β1, hNa_v_β1 or hNa_v_β3 templates. 

In the following we describe two PPI-Id cases to illustrate how Table 1, Table 2 and Appendix A are combined with Figure 2, Figure 3 and Appendix A. Take the upper leftmost corner of Table 1. The value in the cell for PPI-Id 1 of Na_v_1.1α is “Y”, i.e. yes there is a PPI. This corresponds to cysteine (C) in Appendix A (first row entry of data crossed by 3rd col.: “agqCpeg(O)”. This corresponds to the string “agqCpEgym” which was generated by MSA in Table 2. Appendix A informs that C is in contact with arginine (R) of human Na_v_β1 (3rd col. under PPI-Id “1b” the value “R” in string “sckRrse(N)”). For this instance, the spatial configuration is depicted in atomic details (Appendix A). In this case the amide oxygen atom (>C=O) of cysteine forms a hydrogen bond with one nitrogen of β’s arginine. Appendix A informs about this PPI instance in a nongraphical way. The IF has two sides with PPI-Id 1 and PP-Id 1b. On the human pore subunit of isoform 1.1 (hNa_v_1.1α) the string value “agqCpeg(O)” reports that the interacting residue is cysteine (C). It interacts through its backbone oxygen atom (O). On the other side the Appendix A holds the string value “sckRrse(N)” at position PPI-Id “1b” for Na_v_β1. This means that arginine is the counterpart. On an atomic scale nitrogen atom(s) of its monocationic guanidinium head group from its side chain can establish the hydrogen bonding with a strong polar attraction.

The Table 1 in its bottommost rightmost corner informs that no (value “N”) exists for PPI-Id 8 in case of rNa_v_1.9α. Again, Appendix A lends insight not only on a molecular level but also at an atomic scale. The 1.9 isoform’s subunit α has a potential contribution by a cysteine at sequence position PPI-Id 8 (“kehCnss”), but the counter subunits (Na_v_β) present a nonresponding valine (V). This aliphatic residue is conserved on all four β subunits. As a direct result Table 1 summarizes this negative interaction with an “N” qualifier in its corresponding table cell along with all other “Yes” or “No” contributions for the systematic combinations of three Mammalian species, nine isoforms and eight potential PPI sites (labeled PPI-Ids 1 to 8). The two detailed illustrations underline the informative wealth of chemometric studies to complement limited experimental data. They do, however, also raise the molecular modeling practitioner’s challenges concerning management and presentation of data in huge volumes.

#### 2.1.4. Structure Alignment of all Na_v_βs Subunits 

Figure 4 displays the results of the (3D) structure alignment (**SA**), i.e., the ectodomain (IgD) superpositions in space for the 3D templates, namely the cryo-EM structures of eeNa_v_β1, hNa_v_β1 in addition to the crystal structures of hNa_v_β2 to hNa_v_β4 (PDB: 5XSY [48], 6AGF [49], 5FEB [56], 4L1D [55], and 4MZ2 [54]. 

The SA of the subunits eeNa_v_β1 and hNa_v_β1 provides a visual means to identify the interacting residues for eeNa_v_1.4α/eeNa_v_β1 [48] (Figure 2). As can be noticed, essential features remain in close spatial proximity while keeping their properties, except for one of the eight interaction sites: PPI-Id 1 (Figure 4a).

### 2.2. Pore Modulation by Na_v_β Concerning the Acceleration for Fast Gating Inactivation 

Our hypothesis has been based on the displacement of S4 DIII towards the extracellular region during cell membrane depolarization, when it contacts Na_v_β1 or Na_v_β3 (Figure 5). It is not far-fetched to assume that if a voltage sensor gets close enough for noncovalent binding with IgD ectodomains, a spatial rearrangement will take place. Surface charges on Na_v_α will come under the influence of positively charged lysines (K37 on Na_v_β1 or K18 on Na_v_β3, see our Appendix A, cf. Figure 4c,d in [49,55]). Consequently, this displacement could trigger modulation of the pore subunit (Na_v_α). Furthermore, we assume that this contact exert a domain rotation that affects the fast inactivation of Na_v_α gating. 

An electrostatic repulsion could be created with the Na_v_β1 or Na_v_β3 subunits. As a suggested mechanistic consequence, the IF-ECLs move and pull the ECR, which in turn transfers strain energy onto the Na_v_α subunit. This happens precisely on S4 (DI and DIV) and could generate a conformational change in the channel that accelerates fast inactivation and finally closes the modulation cycle with S4 on DI and DIV returning to its starting positions. 

Based on the present findings we propose a possible mechanism of α (pore) modulation by β subunits for the fast inactivation cycle (Figure 6). The acceleration of fast inactivation by the Na_v_β1 or Na_v_β3. In the PPI model, a rotamer library was applied to find favorable Van der Waals contacts [59,60,61] concerning Na_v_α/Na_v_β1 and Na_v_α/Na_v_β3 (Appendix A).

### 2.3. Determination of Relevant ECL Properties 

We quantified all ECL sizes from the nine isoforms for the three species (Appendix A), the ECR properties and those for S5 and S6 ECLs (Appendix A), as well as the ECL properties on the α subunit (Appendix A) [66]. Each isoform and ECL presented inherent characteristics, which were discussed in more detail in Section 3.3.

Moreover, the volume and the solvent-accessible area (SAA)—surface area accessible to solvent—were calculated for each ECL of each isoform (Appendix A) [61].

The polar surface area (PSA), nonpolar surface area (NPSA) and MEPS (negative potential: N-MEPS and positive potential: P-MEPS) were measured for the entire IF-ECL as a grand total and the individual values for each IF-ECL (Appendix A) [61,63,64].

The IF areas of the atoms were determined (Appendix A) besides the buried α and β IF area, i.e., the spatial intersection of IF-ECL with Na_v_β1 and Na_v_β3) (Appendix A) [61]. To get rid of the loop length bias—longer loops tend to possess more chances of interacting residues than shorter loops—we decided to present normalized values, as a general rule: here we took the IF percentage of aforementioned grand total as the 100% basis. The following properties for IF-ECLs with both β subunits were computed: PSA, NPSA, P-MEPS and N-MEPS. In the next step, common treats (similarities) were detected and clustered into the following interaction patterns: S5 DI: (Na_v_1.1α and Na_v_1.3α), (Na_v_1.5α and Na_v_1.7α), (Na_v_1.2α, Na_v_1.4α and Na_v_1.6α), (Na_v_1.8α) and (Na_v_1.9α); S1-S2 DIII: (Na_v_1.1α and Na_v_1.4α), (Na_v_1.3α and Na_v_1.5α), (Na_v_1.2α and Na_v_1.6α), (Na_v_1.8α), and (Na_v_1.9α); S5 DIV: (Na_v_1.1α and Na_v_1.5α), (Na_v_1.2α, Na_v_1.3α and Na_v_1.6α), (Na_v_1.4α and Na_v_1.7α), (Na_v_1.8α), and (Na_v_1.9α). In the case of S6 DIV, however, no similarities have been found among the nine isoforms (Appendix A). Appendix A illustrates the β1 subunit interface and its properties in all details. 

### 2.4. PPI Patterns on Na_v_s Isoforms 

The underpinning of protein functions in most biological processes constitutes the plethora of atom-to-atom interactions between proteins and other biomolecules (cf. interactome). Predicting interactions on an atomic level remains one of the most challenging endeavors in structural biology [67,68,69].

During evolution protein structure is more conserved than its underlying primary sequence, i.e., sequences diverge from a common ancestor but maintain identical or similar functions with little changes due to homologous exchanges at their active sites [70,71,72]. Residues at the sensitive interfaces become significant for geometry or signaling and therefore tend to be conserved in the protein structure [73]. Another well-characterized property of interfaces refers to the existence of “hot spot” residues, which are the residues that make the largest contributions to complex formation [74]. In our study context, several reports have raised the question about what atomic components exactly protein interfaces are made of in order to improve prediction power for PPIs [75].

In vitro research to gain mechanistic insight into Na_v_s at atomic scale has been a daunting task for decades due to its membrane-embedded location and multidomain complexity [6,76]. It is in such situations when chemometric approaches lend insight unraveling hitherto unnoticed atomic patterns.

Na_v_β subunits belong to the immunoglobulin (Ig) superfamily. Its overall structure is an all-beta (strands) fold, which is typical for cell adhesion molecules [77]. Variation in presence (or absence) of β subunits regulates α subunit expression to differentiate tissues. Furthermore, they modulate the pore unit kinetics [78,79] while the mechanism by which this phenomenon occurs has been extensively studied. However, understanding the gating mechanism of the channel and its modulation in details has just emerged in a bitwise manner [80]. Looking back into the channel’s history of research, in 1985 and 2000 respectively, Na_v_β1 and Na_v_β3 were first reported as cell (surface) adhesion proteins in interaction with the α pore unit through noncovalent bonds [81,82]. 

New aspects of subunit cooperativity came from A.P. Jackson’s laboratory with Namadurai et al. in 2014 [55] who have elucidated a trimeric crystal structure of Na_v_β3, i.e., three IgDs in a crystallographic unit cell. Notwithstanding, it has been an unsettled question if this trimer also reflects a biological unit? Recently, molecular dynamics studies indicated that spontaneous oligomerization of a full-length Na_v_β3 subunits to a trimer would probably be a very slow process if it occurred in cell membranes. The three TMH of Na_v_β3 would not interact strongly enough [83]. In addition, our team also analyzed whether the IgDs of the hNa_v_β1 subunits could form trimers [49]. We determined that in the three IgDs strong repulsion exists between the negative total charges of human Asp25 and Glu27 residues upon fitting them onto the spatial positions of each monomers of the Na_v_β3 trimeric structure (PDB: 4L1D [55]). So far, this finding has not been reported elsewhere. Both amino acids have not been exchanged during evaluation across Mammalian species, all of which hints at a pivotal PPI hot spot for Na_v_α subunits (Figure 7, Appendix A). Glass et al. (2020) [83] reasoned that if the hNa_v_β3 trimer were to interact with the VSDs of the pore-forming α protein in analogy to structurally known hNa_v_β1, a substantial rearrangement of the IgDs would be necessary.

Since the crystal structures of α and β1 in complex exist in addition to the trimeric β3 and monomeric β1 (hNavβ3 [55], hNavβ1 [49]) it was possible to carry out structural biology studies by superpositioning them onto each other (Figure 8). Frequently other terms than superpositioning are used: 3D, spatial or structural alignment (**SA**), in addition to fitting or matching (cf. Magic fit under SPDBV or MatchMaker under Chimera). Here, we used hNa_v_1.4α/hNa_v_β1 of the Na_v_β3 trimer [55] and hNa_v_1.4α/hNa_v_β1 [49] (Figure 8 and Figure 9). 

Compared to the sheer number of all-beta fold variations—AKA the Ig superfamily—the ectodomain IgD of hNa_v_β3 [55] is extremely similar to IgD of hNa_v_β1 [49]. Structural evidence concerning eeNa_v_1.4α/eeNa_v_β1 [48], hNa_v_1.4α/hNa_v_β1 [49], and hNa_v_1.7α/hNa_v_β1 [51] was reported about a binding site in the TM region of Na_v_β1 between S1 and S2 helices on VSD DIII [55]. Noteworthy is the finding that the TM regions of the Na_v_β1 and Na_v_β3 subunits possess high sequence similarity. In vitro studies revealed that the TM region of Na_v_β3 non-covalently binds VSD DIII [80] in a way that Na_v_β1 does [48,49,51]. Albeit, there is no structural data to pinpoint the location of the Na_v_β3 binding site on α subunits.

Frequently, it has been assumed that the Na_v_β3 subunit interacts with the Na_v_α subunit through the same mechanism as the Na_v_β1 subunit [84]. Nevertheless, in vitro studies demonstrated that both Na_v_β1 or Na_v_β3 attenuate lidocaine binding to Na_v_1.3α [85]. Said local anesthetic binds the S6 helix of domain IV by noncovalent bonds [86] and structural evidence affirms that Na_v_β1 forms IPP with Na_v_1.4α in the IF-ECL S6 DIV [48].

As working hypothesis, we proposed that the Na_v_β3 binding site on the α pore subunit is the same as that of Na_v_β1. The assumption was based on the structural data analyses of 3D aligned eeNa_v_1.4α/eeNa_v_β1 with hNa_v_1.4α/hNa_v_β1. The same conservation pattern is found again in template complex hNa_v_1.4α/hNa_v_β3 in addition to all isoforms across the three species under scrutiny (Appendix A). 

In the analyzed (3D) template structures and our 3D models we found interaction patterns at the Na_v_α/Na_v_β1 and Na_v_α/Na_v_β3 interfaces. Upon inspection, it is safe to generalize our detailed findings that these interaction patterns significantly diverge between isoforms crossing species. In particular, we noted specific PPI patterns between Na_v_α with either Na_v_β1 or Na_v_β3 subunits.

Applying Chimera’s combined two-dimensional (2D) with 3D alignment capacities for rational protein superposition (by MatchMaker), the templates of hNa_v_β1 [49] and hNa_v_β3 [55] were aligned as spatial references (Figure 4b). Thereupon, we identified all interacting residues between template hNa_v_1.4α/hNa_v_β1 [49] and 3D model hNa_v_1.4α/hNa_v_β3 [49,55] (Figure 3). The identification was assisted by 3D template eeNa_v_1.4α/eeNa_v_β1 [48] as a most valuable reference to pinpoint conservation or homology for closely or far-distantly related organisms, here: three Mammalian species versus eel (Figure 2). As an asset for PPI validation, not only sequential but also structural similarities of hNa_v_β3 with hNa_v_β1 lie significantly above the twilight zone of homology with ≈ 50%, RMSD ≈ 1.2, respectively. Taken together all topology patterns, convincing evidence was unveiled by chemometrics, all of which indicate that hNa_v_β3 subunit binds and modulates Na_v_α from the same position and via the same mechanism as hNa_v_β1 subunit, because the interacting residue pattern is almost the same (see our 3D model of hNa_v_1.4α/Na_v_β3 and PDB entry 6AGF [49] with the hNa_v_1.4α/Na_v_β1 in PDB format in SM). Sufficient(ly tiny) variations do exist, however, on both proteins. They could explain to some degree the differences in channel kinetics which has to be confirmed in future studies with more experimental research. 

For the sake of inference power of our chemometric output data, we also studied the known structures of hNa_v_β2 [56] and hNa_v_β4 [54]. Again, 2D and 3D alignments were carried out with Chimera against reference structure hNa_v_β1 [49]. All critical data clearly lie in the boundaries of the twilight zone with ≈18.8 or 18.1%, RMSD: 4.52 or 7.12, respectively. With hNa_v_β1 [49], hNa_v_β2 [56] (Figure 4c) and hNa_v_β4 [54] (Figure 4d) aligned, the degree of positional mismatches of equivalent residues becomes obvious by eyesight, concerning β2 or β4 sequences **vs** known interacting residues of reference hNa_v_β1 [49]. The loss of conserved positions for the interacting residues of both subunits (β2, β4) strongly hints at the existence of a totally distinct PPI with the α pore subunit. This finding has not been reported in the literature. 

At that stage we can characterize the more general topological behavior of all nine isoforms and lump them together in view of their distinct interaction patterns: (i) isoforms hNa_v_, mNa_v_ and rNa_v_ (1.2, 1.4 and 1.6), present eight PPIs with Na_v_β1 or Na_v_β3 subunits; (ii) isoforms hNa_v_, mNa_v_ and rNa_v_ (1.1, 1.3 and 1.7) present seven PPIs. Both share essential features, so we suggest they have a common effect on modulation. In (weak) contrast, isoforms hNa_v_, mNa_v_, and rNa_v_ (1.1, 1.3, 1.5, 1.7, 1.8, and 1.9) coincide in a non-interacting residue at position 6 in ECL S6 DIV. In stark contrast, isoforms hNa_v_, mNa_v_, and rNa_v_ (1.5, 1.8, and 1.9) always share two features: first, they all lack two or more PPIs and secondly, they all belong to binding type TTX-R, and they associate to an interaction reduction (PPI-Id in Table 1) on S1-S2 DIII ECL, while isoforms hNa_v_, mNa_v_, and rNa_v_ (1.9) do not at all interact through S6 DIV ECL. Wrapping up the findings into a mechanistic picture, gating of all those isoforms belonging to binding type TTX-R might also coincide in a common modulation mechanism. The absence of interaction in PPI-Id 6 in ECL S6 DIV concerns the following isoform: h, m, rNa_v_ (1.1, 1.3, 1.5, 1.7, 1.8, and 1.9). This interaction site (PPI-Id 6) exposes a strong salt bridge for isoforms h, m, rNa_v_ 1.2, 1.4, and 1.6. (Appendix A). We infer that the presence or absence of that strong electrostatic signal at the IF could be a significant feature to trigger isoform-dependent variations in pore modulation. 

## 3. Discussion 

### 3.1. Hypothetical Acceleration of Fast Inactivation of Gating of the Na_v_s 

Cell membrane depolarization is associated with an upwards movement to expose the helical S4 voltage sensors into the cell membrane surface (Figure 6). S4 exposition to the surface implies a conformational change of the channel to enter an open state [19,20,21,22]. As a topological result, the three-amino acid inactivation gate (IFM) located between helices S6 DIII and S1 DIV swiftly connects to the pore and prevents sodium ions to enter (Na^+^ influx), all of which leads to an inactivated state. Thanks to the identification of interacting residues based on our systematic topological (sequential) and structural (spatial) models it is possible to link them to reported electrophysiological aspects about functional transition from activation to inactivation of all nine Na_v_ isoforms (Figure 1) [26,27]. Of note, the variable N-linked glycosylation of the ectodomains (IgDs) does not affect our cheminformatic results because it does not belong to the interface between both subunits [84].

Zhu et al. (2017) [80] concluded from their in vitro studies that Na_v_β1 and Na_v_β3 accelerate the deactivation of S4 in Na_v_1.5, in addition to Na_v_β1 in S4 (DIII and DIV) or Na_v_β3 in S4 (DIII), respectively. In good keeping, Ferrera et al. (2006) [87] demonstrated that Na_v_β1 determines the electrical environment of the channel by changing the surface charges that electrostatically affected the activation of the channels.

In our modeled complexes Na_v_α/Na_v_β1 and Na_v_α/Na_v_β3 (Appendix A), both β subunits are located in proximity to the S4 voltage sensor on DIII. In hypothetical terms, when the cell membrane is depolarized, S4 on DIII shifts in space and subsequently two conserved positively charged residues on S4 on DIII enter into repulsive contacts with a conserved lysine on both β subunits (Appendix A). All three positive charges are clearly exposed on the solvent-accessible area on the protein surface (Figure 5). It is safe to conclude that a strong biochemical signal is triggered when this charge repulsion takes place at S4 DIII (Figure 8). As a direct consequence for the channel’s overall geometry, domain shift of IgDs (the ectodomains of Na_v_β1 and Na_v_β3) takes place to modulate fast inactivation mechanism. The segment S1–S2 on DIII serves as a flexible hinge to form a noncovalent association with β1 and β3 IgDs. In a slower response to depolarization, S5 on DI and S5 on DIV terminate the cycle by forcing S4 (DI and DIV) back into its initial position in a spring-like fashion. Finally, S6 on DIV arrests—like an anchor—both β subunits. On theoretical ground, the present findings explain the cooperativity between ECRs and certain Na_v_β subunits for channel pore modulation. From an evolutionary point-of-view, this makes sense, since calling-in external proteins (auxiliary βs) to assist the protein function (α pore) is achieved much faster than the adaption of loop segments by slow selection of random point mutations over time. Obviously, this happened at a time during cellular evolution when gene fusion already took place to transform the homotetrameric (bacterial) channel into a single chain pore protein, which is composed of four different domains (“monocadenar hetero-tetra-domain subunit”). In this context the existence of nine closely related mutants (isoforms) show the work of evolution from the near-past to present time when a common ancestral protein has been evolving along with differential gene expression and tissue specialization. 

### 3.2. Properties of the ECL Residues for Na_v_α Subunits

The ECL sequence lengths are shown in Appendix A. Intriguingly, S5 on DI constitutes the longest ECL. Its aberrant size reflects that it encompasses a second PPI; PPI-Id 1 is conserved in every isoform and species; PPI-Id 2 is conserved in almost all isoforms with the exception of Na_v_1.8α and Na_v_1.9α. The ECL length of S5 on DIV contains a negatively charged residue (Id 5) which is conserved throughout all nine isoforms for all three species. Possibly, the ECLs of S5 on DI and DIV have evolved steadily in contact with Na_v_β1 and Na_v_β3 subunits, leading to speculations about their role as main binding sites for Na_v_α modulation.

The chemometric properties describing S5 and S6 are documented in Appendix A. In all three species Na_v_1.4α (resp. Na_v_1.5α isoform) accounts for the highest (respectively, lowest) amount of polar and negatively charged residues on S5 and S6. Two isoforms contain the least number of nonpolar residues, namely Na_v_1.8α and Na_v_1.9α. Both show the highest percentage of polar ECR residues and in particular on ECL S5 and S6 for all three species. Our finding here reflects the extant literature speculating about the absence of experimental evidence for Na_v_1.9α cooperation with β subunits for gating. Appendix A summaries the ECL properties. Na_v_1.4α and Na_v_1.6α possess the same number of PPIs. Despite this common treat, subtle differences may also explain why their kinetic behavior differs. The ECL of S5 on DI in Na_v_1.4α (resp. Na_v_1.6α) hosts the highest (resp. smallest) amount of polar and negatively charged amino acids. Intriguingly, the Na_v_1.6α isoform accumulates even more polar and negatively charged residues in its S5 loops on DIII. Domains DI and DIII are facing each other from diametrically opposing positions across the central pore part. Both have developed the longest S5 ECLs among all Na_v_α. It can be speculated that their lengths could reflect the main electrostatic attraction of the Na_v_α for the conduction of Na^+^ towards the channel pore. This balance of residues distributed in the S5 ECLs (DI and DIII) probably has evolved to provide the Na_v_1.4α and Na_v_1.6α isoforms a MEPS similar to a fingerprint with inherent characteristics to perform a specific function on the tissue. Aforementioned findings have not been reported in the extant literature that far.

### 3.3. Volume and Surface Properties of the ECL on Na_v_α Subunits

Evolution leads to random point mutations or SNPs with variable consequences for survival of organisms. On molecular level it changes structures and functions [88,89]. Isoforms can be understood as transient states during divergent evolution to separate them from a common ancestral protein when cells evolve to more specialized tissues in organisms [90,91]. Biochemical signaling is not seldom located on exposed loop segments on cell surfaces with a remarkable conservation of signal-relevant residues amidst the variable loop segments. This observation is the rationale to combine 2D and 3D alignment techniques enabling us to reveal this hidden world of signaling or interacting amino acids at the α/β interface [92,93,94]. The type of protein structure—AKA fold unit—is more conserved than its underlying primary sequence. Moreover, unchanged structures keep the biochemical function, what sometimes can be observed even in extreme cases of sequence divergence [95,96,97]. Of note, each ECR has 16 ECLs on each isoform.

With respect to all nine isoforms, the Na_v_1.4α isoform has the largest volume and widest SAA concerning the ECR in general. Moreover, regarding the pore architecture, this holds true also for S5 and S6 of ECLs. In contrast, the Na_v_1.9α isoform it has the smallest volume and SAA in the ECR and ECLs S5 and S6 (Appendix A). This finding has not yet been reported by others. 

Appendix A, respectively, display the molecular volume and SAA of the ECLs. The S5 DI ECLs on h, m, rNa_v_1.4α have a fairly larger molecular volume than the other isoforms. In contrast, the molecular volume of the S6 DIV ECLs on Na_v_1.9α is significantly smaller than on all other isoforms. This finding nicely explains why Na_v_1.9α isoforms do not enter in contact both β subunits via ECL S6 DIV and has not yet been reported by others either.

Of all isoforms, the h, m, rNa_v_1.4α (h, m, rNa_v_1.8α and h, m, rNa_v_1.9α) isoforms have the highest N-MEPS (P-MEPS) in the IF-ECLs (Appendix A). These patterns are identical for S5 DI and S6 DIV ECLs. On the other hand, Na_v_1.8α and Na_v_1.9α have higher P-MEPS for S1-S2 DIII ECLs. Intriguingly, Na_v_1.4 isoforms possess the smallest areas of N-MEPs in S1-S2 DIII ECLs (Appendix A). The isoform-dependent characteristic features of each isoform could be attributed to the affinity of electrostatic attraction to the Na_v_β1 and Na_v_β3 subunits.

### 3.4. Interface Properties Between Na_v_α and Na_v_β1 or Na_v_β3

The observation that only a tiny portion of the total surface area belongs to structural or functional segments is reflected by high conservation at those segments [98,99,100,101]. Especially electrostatic forces often act as critical determinants for biochemical signaling or other protein functions like ligand recognition, affinities, or structural stability. PPI is said to take place at surface locations (patches) with geometric and chemical complementarity [101,102,103,104,105]. This way, ECLs on the sodium channel tend to keep physicochemical similarities on their surfaces all of which sum up into distinct interaction patterns.

The variable surface area between the IF-ECLs and both β subunits was documented in Appendix A. For most of the Na_v_s isoforms, the following general interaction pattern holds in order of shrinking surface: S5 DI > S1–S2 DIII > S6 DIV > S5 DIV, with the exception of Na_v_1.9 isoforms, where the interface area of the IF-ECL S6 DIV and the Na_v_β1 and Na_v_β3 subunits was found to be much smaller compared to all other isoforms. The observation is in excellent keeping with the electrophysiological role of h, m, rNa_v_1.9α, lacking PPI with both β subunits. Hence, it seems not far-fetched to infer that it does not interact with IF-ECL S6 DIV.

Appendix A inform about the percentage scores concerning IF-ECLs and both β subunits for the following properties: PSA, NPSA, P-MEPS, and N-MEPS, measured on an atomic scale, which only counts the PPI atoms (Appendix A). The isoforms Na_v_1.2 and Na_v_1.6 possess interaction patterns, which resemble those of the IF- ECLs S5 DI, S1-S2 DIII, and S5 DIV. Our finding here could explain why the interface surface properties in both subunits (Na_v_α and Na_v_β) are conserved because the reflect similar modulation. Interestingly, Na_v_1.2α and Na_v_1.6α present the same PPI sites along with Na_v_1.4α. Yet, the size, residue properties, SAA, molecular volume, differ greatly between either Na_v_1.2α or Na_v_1.6α versus Na_v_1.4α, emphasizing the IF-ECLs on S5 DI. On the other hand, IF-ECLs on S6 DIV do not show any similarity between isoforms, all of which is in line with the pivotal role of IF-ECLs S6 DIV as a strong contributor to pore modulation because of its unique electrostatic forces on its surface.

## 4. Materials and Methods 

### 4.1. In Silico Homology Modeling

Homology modeling techniques were carried out to generate (3D) structure models of the hitherto unknown β subunits of mice and rats (mNa_v_β1, mNa_v_β3, rNa_v_β1, and rNa_v_β3) as well as the isoforms of the following Na_v_α/Na_v_β complexes: hNa_v_1.1α, hNa_v_1.3α, hNa_v_1.6α, hNa_v_1.8α, hNa_v_1.9α, mNa_v_1.1α to mNa_v_1.9α, rNa_v_1.1α to rNa_v_1.4α, rNa_v_1.6α to rNa_v_1.9α. Two programs, MODELLER 9.22 [106] and Chimera alpha V.1.14 [61], were applied using the following cryo-EM as (3D) templates: hNa_v_1.4α/hNa_v_β1 with chains A and B from PDB entry 6AGF [49], hNa_v_1.2α with chain A from PDB entry 6J8E [50], hNa_v_1.7α with chain A, from PDB entry 6J8G [51], rNa_v_1.5α from PDB entry 6UZ0 [52]. The following crystal structures were also taken as templates: hNa_v_β2 from PDB entry 5FEB [56], hNa_v_β3 with chain A from PDB entry 4L1D [55], and finally hNa_v_β4 from PDB entry 4MZ2 [54]. 

### 4.2. Identification of Interacting Residues at the Interface Between Na_v_α and Na_v_β

The advent of a complete sodium channel structure with pore part and auxiliary proteins from a higher (vertebrate) organism has ushered a new area of structural biology analysis to lend insight into the underpinnings of subunit modulation mechanisms. Th structure elucidation was a pivotal step because vertebrate Na_v_ channels are composed of a monomeric (single-chain) α protein with four different domains (I to IV), in contrast to the hitherto known homo-tetrameric (4 chains) channels of bacterial species without auxiliary proteins. Precisely, our cheminformatic work exploits this first experimentally observed interface of Na_v_1.4α isoform complex from a vertebrate species: the eel (eeNa_v_α/Na_v_β for short, PDB entry: 6AGF [48]). This template helped analyze the interacting amino acids which form the PPI in the ECR (here: eeNa_v_α/Na_v_β) applying software tools to detect contacts at the interface and the databases of Dunbrack and Dynameomics rotamers [59,60]. MSA was always performed using web based Clustal Omega 1.2.4 under its default settings [62]. To get rid of the residue numbering problems in sight of variable sequence lengths, each identified residue as “interacting” (i.e., forming the PPI of eeNa_v_1.4 α/eeNa_v_β1 [48]) was labelled as a small segment of seven adjacent amino acids. In its central position, the interacting residue is flanked by three amino acids on either side. The schematic pattern is “yyyXyyy”, where “y” symbolizes any amino acid, while “X” is the interacting amino acid.

### 4.3. The Input Structures as Templates for the PPI Models

To generate the interfaces the following templates were used: the cryo-EM structure of hNavβ1 from PDB entry 6AGF [49], chain A of hNa_v_β3 crystal structure from PDB entry 4L1D [55]. Moreover, the following homology models were generated for the mNa_v_β1 subunits, mNa_v_β3, rNa_v_β1, and rNa_v_β3, in addition to the computed isoform complexes hNa_v_1.1α, hNa_v_1.3α, hNa_v_1.5α, hNa_v_1.6α, hNa_v_1.8α, hNa_v_1.9α, mNa_v_1.1α to mNa_v_1.9α, rNa_v_1.1α to rNa_v_1.4α, rNa_v_1.6α to rNa_v_1.9α. We also used the cryo-EM structures of hNa_v_1.4α isoform with chains A and B from PDB entry 6AGF [49]), hNa_v_1.2α with chain A from PDB entry 6J8E [50]), hNa_v_1.7α with chain A from PDB 6J8G [51]), rNa_v_1.5α from PDB entry 6UZ0 [52]). Of note, no chimeric combinations were made crossing species.

### 4.4. Determination of the Extracellular Regions of all Na_v_α Subunits

The structures and sequences of the ECLs of the isoforms hNa_v_1.1α to hNa_v_1.9α, mNa_v_1.1α to mNa_v_1.9α and rNa_v_1.1α to rNa_v_1.9α, were determined, based on the available template structures eeNa_v_1.4α, hNa_v_1.2α, hNa_v_1.4α, rNa_v_1.5α and hNa_v_1.7α which had been downloaded from the Orientations of Proteins in Membranes (OPM) database [107]. OPM provides spatial information about the lipid bilayer packing of the transmembrane helical part for our channel models. OPM helped define the ECR, i.e., at which position the loop protrudes and re-enters TMH.

### 4.5. Calculation of the Properties for all ECLs 

The 432 amino acid segments, which define all ECLs under scrutiny were computed, i.e., D1 to DIV with 4 ectodomain loops, and each of them by nine isoforms for three species yield 16 × 9 × 3 = 432 topological models. They were “extracted” from the primary sequences of hNa_v_1.1α to hNa_v_1.9α, mNa_v_1.1α to mNa_v_1.9α and rNa_v_1.1α to rNa_v_1.9α. The ECL lengths and informative properties about the interacting amino acids were systematically computed and documented for subsequent PPI analyses. The plethora of data made scripting a most valuable asset under Chimera (see scripts at the end of SM). Chemometric properties—AKA descriptors or parameters—included polar and nonpolar area, cysteines, or aromatic amino acids. Of note, hydrogen bonding was observed and documented, but data presentation omitted, since constructing the intra- or intermolecular hydrogen networking is a standard option. The H-bonds at interfaces are readily on display, i.e., intermolecular hydrogen networks (in the contact zone, which is defined by an atom selection radius) between two proteins (see our 3D model of hNa_v_1.4α/Na_v_β3 and PDB entry 6AGF [49] with the hNa_v_1.4α/Na_v_β1 in PDB format in SM). 

### 4.6. Calculation of the Chemical Surface Properties for All IF-ECLs

For all 3D models the potential energies of the structures were minimized and total and partial charges loaded with a water probe radius of 1.4 Å and a vertex density of 2.0 [108]. The molecular volume, total SAA as well as the polar or non-polar SAA of charged or uncharged atoms were estimated in Å^2^ for all ECLs [61]. As usual all data was computed for all nine isoforms of the three species. 

### 4.7. Electrostatic Interactions of the IF-ECLs Surfaces (MEPS) 

Structural input files (models and 3D templates) were prepared with Chimera add-on PDB2PQR, according to a protocol [63]. It computes Poisson-Boltzmann electrostatics, which constitutes a higher level of theory than electrostatic forces calculated based on Poisson–Boltzmann equation solver (APBS) [64]. The numerical output was converted for graphical display of MEPS. To this end, all those data points above and below a given threshold (±30) were excluded (empiric protocol). Electrostatic force values in the +30 to −30 range were considered for linear scaling the color code between +1 and −1 (unit one data normalization). The corresponding load at each vertex of the surface was assigned using the UCSF Chimera alpha V. 1.14 interface [61] in molecular selections for each ECL. 

### 4.8. Calculation of the ECL Surface Properties at the Interface with Na_v_β1 and Na_v_β3 Subunits 

At the interface the buried area, total SAA and polar and non-polar areas were calculated in Å^2^ applying scripts under Chimera [61]. For direct comparison some values were expressed as percentages to reflect the relative portion (%) of the total loop length (100 %) to account for the huge variation in length. In addition, residues with positive or negative charges were taken as basis to compare MEPS at the interface. 

### 4.9. External Model Validation of the PPI Models

In a more general view, the advent of structural knowledge about the cell proteome has ushered a new area of PPI studies identifying hotspots of interaction between adjacent proteins [109]. After finishing our study, a fully automated interface generation was carried out. The web-based tool identified the same interacting residues (see final section of the Appendix A [110]).

## 5. Conclusions 

In this work, we analyzed observed protein-protein interfaces and postulated others in models derived from experimentally determined 3D templates. The models were generated for all nine existing isoforms of three species (human, rat, mouse). The interface concerned the residues of extracellular loops in close contact with the Na_v_β1 or Na_v_β3 subunits. Thanks to the chemometric analysis, we formulated a model for fast inactivation of the Na_v_α pore gating modulated by the presence of either Na_v_β1 or Na_v_β3 auxiliary proteins. On theoretical ground, we gained mechanistic insight of the movements around the S4 DIII voltage sensor, which is modulated by β subunits.

We describe the modulation of the sodium channel activity in terms of a schematic PPI model between pore-containing transmembrane α protein and the auxiliary β proteins for all nine isoforms in three Mammalian species. Our structural models and topological analysis of sequences lead to the conclusion that their distinct interfaces reflect the observed differences in gating kinetics.

We computed chemometric patterns for criteria like non-covalent bonding, loop length, area or volume, solvent accessible area or buried surfaces and other electrostatic descriptors. Isoforms were grouped together according to common interaction patterns and opposed to others with different patterns, and all results were mechanistically related to reports on gating kinetics. The patterns included solvent accessible area or conserved positions for opposingly charged residues on either side of the interface. Our findings about subtle variations in the electrostatic patterns affect the individual modulation capacity of each isoform all of which is in keeping with electrophysiologic observations of gating kinetics and graphically resumed in our schematic drawings.

Our cheminformatic study was thoroughly based on observations taken from the extant literature, and our results are in line with their experimental findings. In addition, we report two hitherto unidentified interaction patterns (or patches) for the 3D templates as well as the proposed interface models. They fit into a larger mechanistic picture with the other interaction patches, which were first reported by Yan et al. with the advent of a complete sodium channel structure for vertebrate species (eel) [48].

This work could orient future research in molecular biology or help design site-directed mutagenesis studies at the subunit interface of voltage-gated sodium channels. In particular, molecular dynamics studies on supercomputers could simulate the gating trajectories over time and confirm that the isoform movements can be grouped together following the proposed cheminformatic patterns.

## Figures and Tables

**Figure 1 molecules-25-03551-f001:**
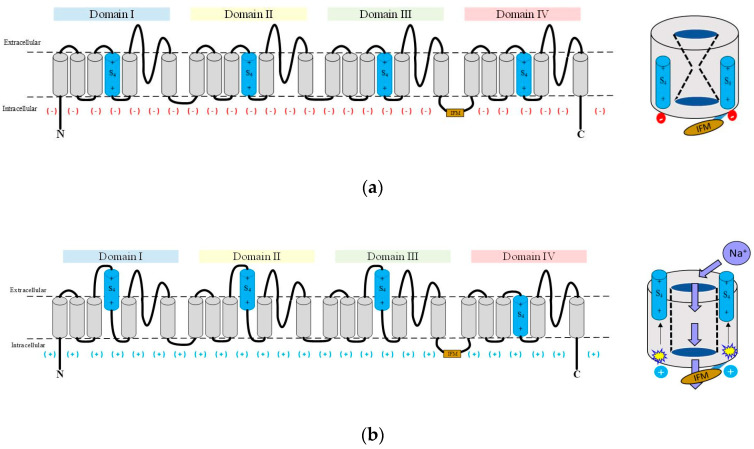
Schematic representation of gating. The three schemes of eukaryotic Na_v_s show the (**a**) closed; (**b**) open; and (**c**) inactivated gating states. (**d**) A typical membrane current of *Rattus norvegicus* of the Na_v_α1.4 isoform responds to a depolarizing pulse reflecting the three main states of gating; IFM inactivation gate: brownish; S4 voltage sensors: sky blue.

**Figure 2 molecules-25-03551-f002:**
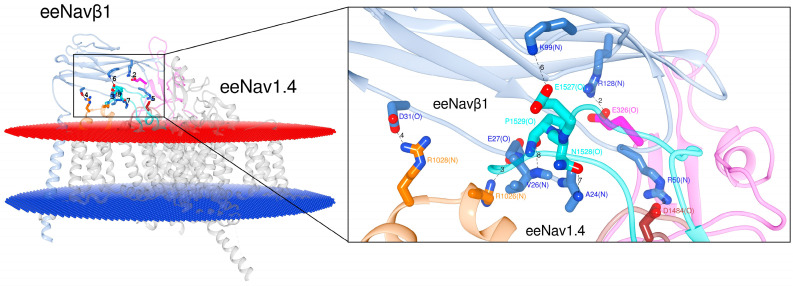
Display of the 3D model for S6 DIV in eeNa_v_1.4α/eeNa_v_β1 [48]. The box displays details about the interacting residues at the interface. Labels 2 to 8: PPI identification numbers (PPI-Ids) of computed polar interactions (Appendix A); the amino acids are labeled by one-letter-codes with their primary sequence residue numbers and interacting atoms, e.g. A24(N). Colors: extracellular membrane boundaries (dark red); intracellular membrane boundaries (navy blue); transmembrane and intracellular protein regions of Na_v_α which do not participate in PPI (gray); Na_v_β1 subunit (cornflower blue); S5 DI: magenta; S1-S2 DIII: orange; S5 DIV: brown; S6 DIV: cyan; computed polar interactions: black dotted lines. Visualization achieved by Chimera Alpha 1.14.

**Figure 3 molecules-25-03551-f003:**
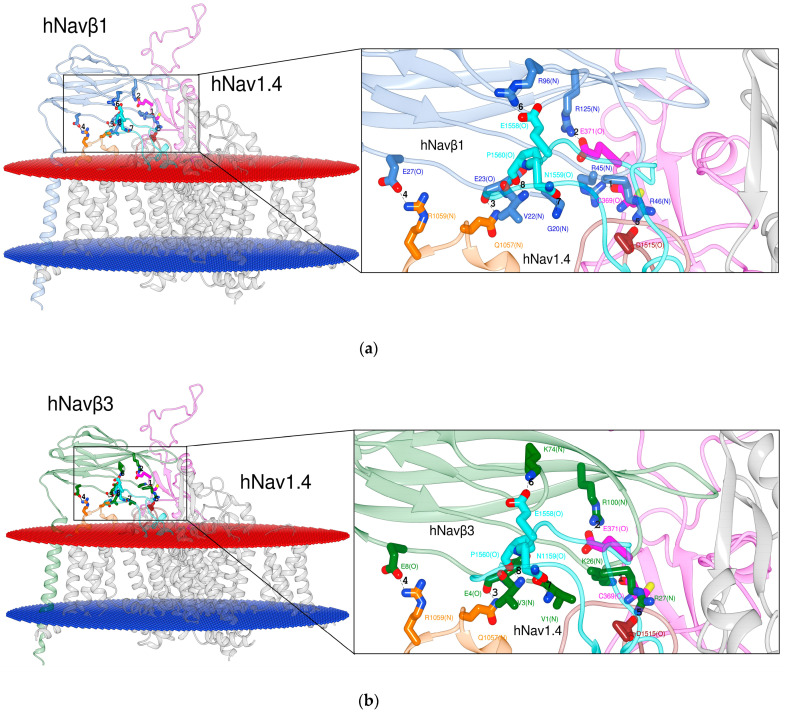
Display of PPI models. Based on the 3D template in (**a**) hNa_v_1.4α/hNa_v_β1 [49]; based on a homology model in (**b**) hNa_v_1.4α/hNa_v_β3 [49,55] for S6 DIV. The box presents atomic details at the interface. Labels 1 to 8: Ids of computed polar interactions (Appendix A); the amino acids are labeled by one-letter-codes with their primary sequence residue numbers and interacting atoms in parentheses, e.g. bottommost D1515(O). Colors: extracellular membrane boundaries (dark red); intracellular membrane boundaries (navy blue); transmembrane and intracellular protein regions of Na_v_α that do not participate in PPI (gray); Na_v_β1 subunit (cornflower blue); Na_v_β3 subunit (forest green); S5 DI: magenta; S1-S2 DIII: orange; S5 DIV: brown; S6 DIV: cyan); computed polar interactions: black dotted lines. Visualization achieved by Chimera Alpha 1.14 [61].

**Figure 4 molecules-25-03551-f004:**
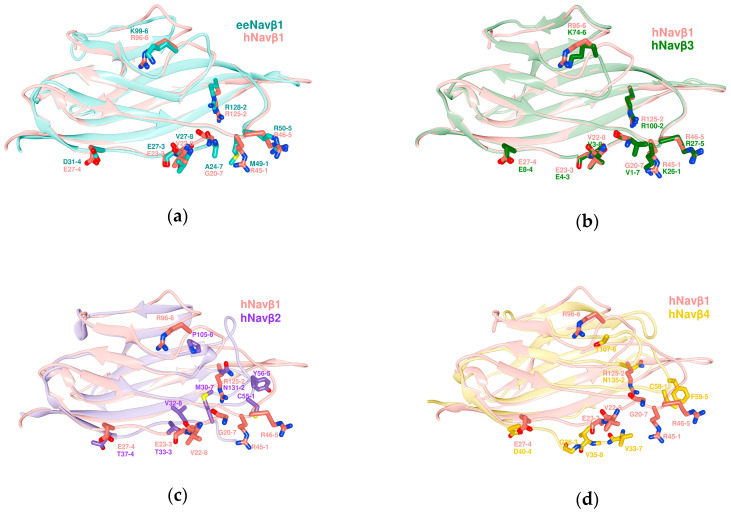
Structure alignments of the ectodomains (IgD) of β subunit templates. (**a**) eeNa_v_β1 with hNa_v_β1; (**b**) hNa_v_β1 with hNa_v_β3; (**c**) hNa_v_β1 with hNa_v_β2; (**d**) hNa_v_β1 with hNa_v_β4; eeNa_v_β1: sea green; hNa_v_β1: salmon; hNa_v_β3: forest green; hNa_v_β2: purple; hNa_v_β4: golden; labels 1 to 8: positions of MSA residues according to Table 2. All superpositions were achieved by Chimera Alpha 1.14 with MatchMaker [61].

**Figure 5 molecules-25-03551-f005:**
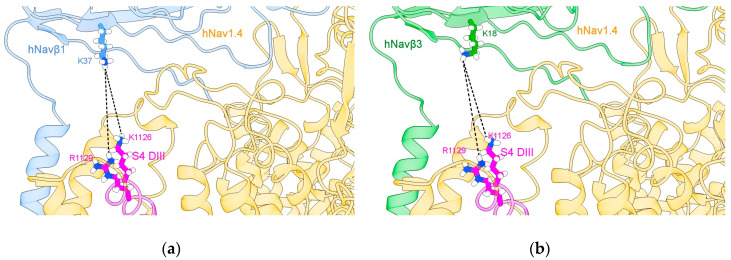
S4 DIII voltage sensor of hNa_v_1.4α [49] in close contact with hNa_v_β1 [49] or hNa_v_β3 [55]. (**a**) hNa_v_1.4α interfaced with hNa_v_β1; (**b**) hNa_v_1.4α interfaced with Na_v_β3; (**c**,**e**) the MEPS at the interface hNa_v_1.4α/hNa_v_β1; (**d**,**f**) the MEPS at the interface between hNa_v_1.4α and hNa_v_β3. The structures were prepared with Chimera add-on PDB2PQR [63] and MEPS calculated for PPI surfaces using the Adaptive Poisson-Boltzmann Solver (APBS) [64], a plug-in tool in Chimera Alpha 1.14 [61] and simulated under Chimera X [65]. S4 DIII voltage sensor: magenta; hNa_v_β1: cornflower blue; hNa_v_β3: green.

**Figure 6 molecules-25-03551-f006:**
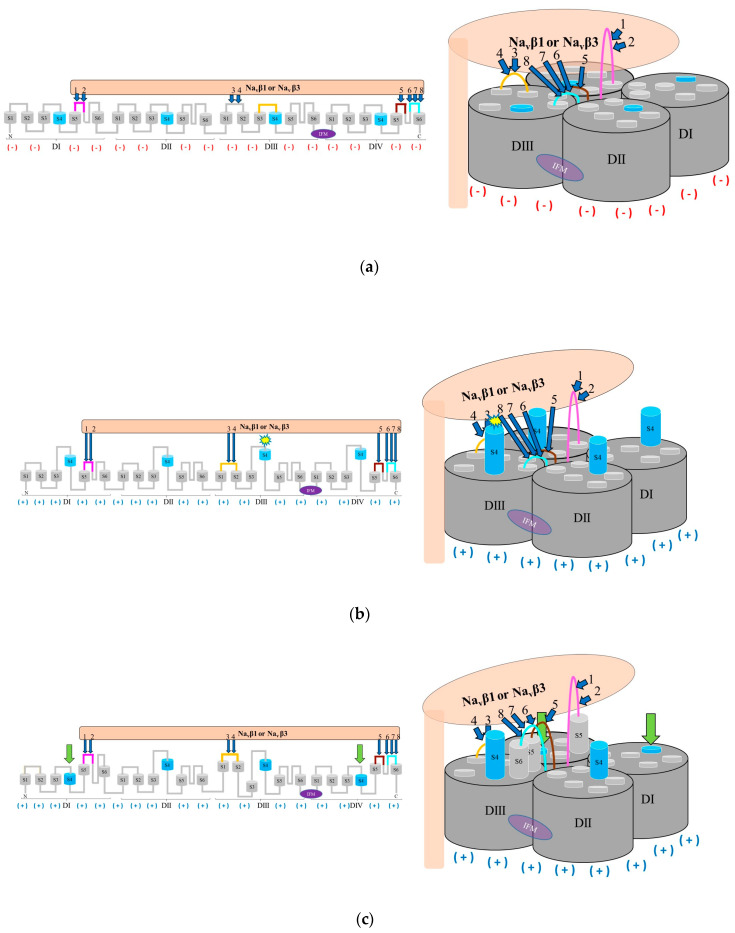
Hypothetical modulation of fast inactivation of Na_v_α gating by Na_v_β1 or Na_v_β3; (**a**) Na_v_α in idle (closed) state in response to interacting Na_v_β1 or Na_v_β3; (**b**) Na_v_α in open (activated) state in presence of Na_v_β1 or Na_v_β3 modulation; (**c**) fast inactivation modulated by Na_v_β1 or Na_v_β3; (**d**) fast inactivation triggered by the IFM inactivation gate; labels 1 to 8: Id of computed polar PPIs (Appendix A); computed polar PPIs: navy blue arrows; return to its start position of S4 is forced by IF-ECLs by computed polar interactions with Na_v_β1 or Na_v_β3: green arrows; negative charges: red minus signs in parentheses; positive charges: navy blue plus signs in parentheses; Na_v_α regions without PPI: dark and light gray; Na_v_β1 or Na_v_β3 subunit: light salmon; S5 DI: magenta; S1-S2 DIII: orange; S5 DIV: brown; S6 DIV: cyan; S4: sky blue, segments S1, S2, S3, S4, S5, and S6: light gray.

**Figure 7 molecules-25-03551-f007:**
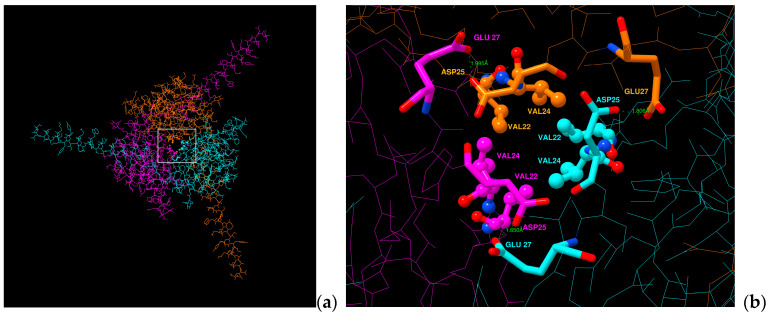
Display of a trimeric hNa_v_β1 model. (**a**) Na_v_β1 trimer seen top-down (**b**) close-up view from above of the alleged hotspot (**c**) Aligned sequences from Mammalian species. Negatively charged residues are identical to Asp25 and Glu27 on Na_v_β1 from *Homo sapiens*; white box: analysis area; green dotted lines: computed repulsion of charges; sticks: negative charged residues; sticks and balls: residues that form a possible hydrophobic patch. Data generated by Chimera Alpha 1.14 [61].

**Figure 8 molecules-25-03551-f008:**
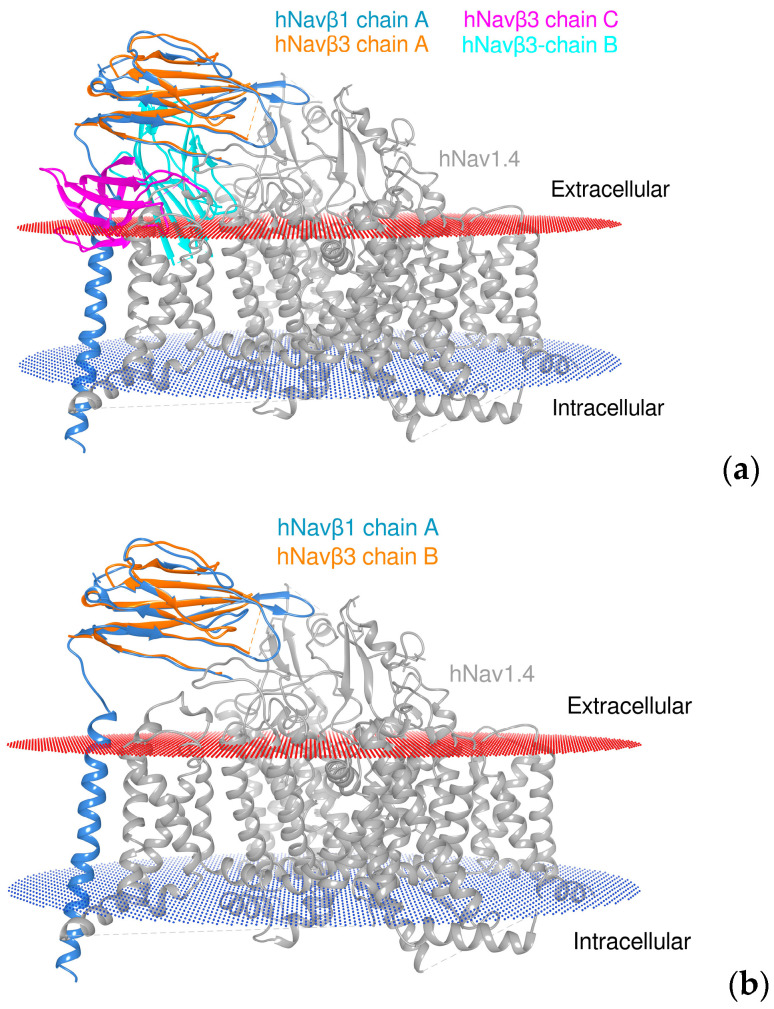
Superposition of monomeric and trimeric 3D models of β proteins. (**a**) Experimentally determined homotrimeric hNa_v_β3 [55] (three colors: magenta, light blue, beige) in superposition with template hNa_v_1.4/hNa_v_β1 (bluish/grey) [49] and (**b**) one IgD (out of three) subunit(s) of homotrimeric hNa_v_β3 (beige) in superposition with template hNa_v_1.4α/hNa_v_β1. It can be seen—by eyesight—that in case a two out of the three subunits bump into the membrane. Extracellular membrane boundaries: dark red; intracellular membrane boundaries: navy blue; Na_v_α subunit: gray; hNa_v_β1 subunit: cornflower blue; hNa_v_β3 chains A, B, and C: orange, cyan, and magenta, respectively.

**Figure 9 molecules-25-03551-f009:**
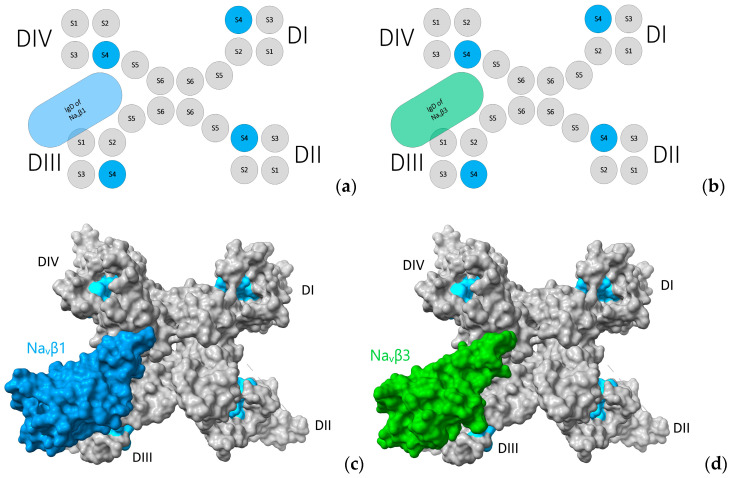
Three-dimensional (3D) location of subunits Na_v_β1 and Na_v_β3. (**a**,**b**) Na_v_α topology in complex with Na_v_β1 and Na_v_β3. Display of 3D models with solvent-excluded surface areas in panels (**c**,**d**). (**c**) Cryo-EM structure of hNa_v_α1.4 in complex with Na_v_β1 [49] and (**d**) Cryo-EM structures of hNa_v_α1.4 [49] in complex with crystal structure hNa_v_β3 [55] positioned according to structural analysis; S4: sky blue; in panels (**c**,**d**) the molecular surfaces are colored: Na_v_β1: cornflower blue; Na_v_β3: green forest; and grey color for Na_v_α subunit surfaces. The same colors were applied to the panels (**a**,**b**) above. 3D models by Chimera X [65].

**Table 1 molecules-25-03551-t001:** Synopsis of the studied PPI patterns. The interaction sites are labeled as **PPI-Id** with Arabic numerals from “1” to “8” to identify them. The resulting patterns are labeled with Roman numerals from “I” to “IX”. Certain PPI between Na_v_α/Na_v_β1 and Na_v_α/Na_v_β3 have been observed experimentally by structure elucidation. Their respective PDB entries and PPI-Id values are marked in bold face. They were used as templates for the reminder. The Y/N values in the table cells symbolize YES/NO referring to the presence / absence of contributions to PPI.

Isoform	IF-ECLs ^6^	S5 DI		S1-S2 DIII		S5 DIV	S6 DIV			PPI Pattern
	PPI-Id ^3^	1	2	3	4	5	6	7	8	
hNa_v_1.1α ^5^	P35498 ^2^	Y	Y	Y	Y	Y	N	Y	Y	I
mNa_v_1.1α ^5^	A2APX8 ^2^	Y	Y	Y	Y	Y	N	Y	Y	I
rNa_v_1.1α ^5^	P04774 ^2^	Y	Y	Y	Y	Y	N	Y	Y	I
hNa_v_1.2α ^4^	**6J8E** ^1^	**Y**	**Y**	**Y**	**Y**	**Y**	**Y**	**Y**	**Y**	**II**
mNa_v_1.2α ^5^	B1AWN6 ^2^	Y	Y	Y	Y	Y	Y	Y	Y	II
rNa_v_1.2α ^5^	P04775 ^2^	Y	Y	Y	Y	Y	Y	Y	Y	II
hNa_v_1.3α ^5^	Q9NY46 ^2^	Y	Y	Y	Y	Y	N	Y	Y	I
mNa_v_1.3α ^5^	A2ASI5 ^2^	Y	Y	Y	Y	Y	N	Y	Y	I
rNa_v_1.3α ^5^	P08104 ^2^	Y	Y	Y	Y	Y	N	Y	Y	I
eeNa_v_1.4α ^4^	**5XSY** ^1^	**N**	**Y**	**Y**	**Y**	**Y**	**Y**	**Y**	**Y**	**IX**
hNa_v_1.4α ^4^	**6AGF** ^1^	**Y**	**Y**	**Y**	**Y**	**Y**	**Y**	**Y**	**Y**	**II**
mNa_v_1.4α ^5^	Q9ER60 ^2^	Y	Y	Y	Y	Y	Y	Y	Y	II
rNa_v_1.4α ^5^	P15390 ^2^	Y	Y	Y	Y	Y	Y	Y	Y	II
hNa_v_1.5α ^5^	Q14524 ^2^	Y	Y	N	Y	Y	N	Y	Y	III
mNa_v_1.5α ^5^	Q9JJV9 ^2^	Y	Y	N	Y	Y	N	Y	Y	III
rNa_v_1.5α ^4^	**6U70** ^1^	**Y**	**Y**	**N**	**Y**	**Y**	**N**	**Y**	**Y**	**III**
hNa_v_1.6α ^5^	Q9UQD0 ^2^	Y	Y	Y	Y	Y	Y	Y	Y	II
mNa_v_1.6α ^5^	Q9WTU3 ^2^	Y	Y	Y	Y	Y	Y	Y	Y	II
rNa_v_1.6α ^5^	O88420 ^2^	Y	Y	Y	Y	Y	Y	Y	Y	II
hNa_v_1.7α ^4^	**6J8G** ^1^	**Y**	**Y**	**Y**	**Y**	**Y**	**N**	**Y**	**Y**	**I**
mNa_v_1.7α ^5^	Q62205 ^2^	Y	Y	Y	Y	Y	N	Y	Y	I
rNa_v_1.7α ^5^	O08562 ^2^	Y	Y	Y	Y	Y	N	Y	Y	I
hNa_v_1.8α ^5^	Q9Y5Y9 ^2^	Y	Y	Y	N	Y	N	Y	Y	IV
mNa_v_1.8α ^5^	Q6QIY3 ^2^	Y	Y	N	N	Y	N	Y	Y	V
rNa_v_1.8α ^5^	Q62968 ^2^	Y	N	N	N	Y	N	Y	Y	VI
hNa_v_1.9α ^5^	Q9UI33 ^2^	Y	N	Y	N	Y	N	N	N	VII
mNa_v_1.9α ^5^	Q9R053 ^2^	Y	Y	N	N	Y	N	N	N	VIII
rNa_v_1.9α ^5^	O88457 ^2^	Y	Y	N	N	Y	N	N	N	VIII

^1^ PDB entry (http://www.rcsb.org/); ^2^ UniProt code (https://www.uniprot.org/); ^3^ PPI-Id for computed polar interactions between hNa_v_α and hNa_v_β subunits (Appendix A); ^4^ homology-modeled and refined structures (3D templates), see Appendix A; ^5^ models; ^6^ ECLs which form interfaces with Na_v_β1 or Na_v_β3 are called IF-ECLs; Y: Interaction; N: No interaction; eeNa_v_1.4α: *Electrophorus electricus* Na_v_1.4α isoform; labels I to XI: Groups of PPI patterns.

**Table 2 molecules-25-03551-t002:** Multiple sequence alignments for either all nine α subunits or all four β subunits of three Mammalian organisms in addition to eel 3D template. MSA identified eight conserved or homologous residues on Na_v_α and Na_v_β. Asterisks (*) in first column: 3D template structures from PDB. Capital letters in bold face: the eight residues. Lower case letters: amino acid neighbors of the eight residues for numberless identification. Colors: positively and negatively charged residues in blue and red, respectively; polar or non-polar residues in cyan or orange.

Isoform	UniProt Code ^1^	ECL			
		S5 DI	S1-S2 DIII	S5 DIV	S6 DIV
		1, 2	3, 4	5	6, 7, 8
hNa_v_1.1α	P35498	agq**C**p**E**gym	yid**Q**r**K**tik	gid**D**mfn	pnk**VNP**gss
mNa_v_1.1α	A2APX8	agq**C**p**E**gym	yid**Q**r**K**tik	gid**D**mfn	pnk**VNP**gss
rNa_v_1.1α	P04774	agq**C**p**E**gym	yid**Q**r**K**tik	gid**D**mfn	pnk**VNP**gss
hNa_v_1.2α	Q99250	agq**C**p**E**gyi	yie**Q**r**K**tik	gid**D**mfn	pdk****DHP****gss
mNa_v_1.2α	B1AWN6	agq**C**p**E**gyi	yie**Q**r**K**tikd	gid**D**mfn	pek****DHP****gss
rNa_v_1.2α	P04775	agq**C**p**E**gyi	yie**Q**r**K**tik	gid**D**mfn	pek****DHP****gss
hNa_v_1.3α	Q9NY46	agq**C**p**E**gyi	yie**Q**r**K**tik	gid**D**mfn	pdt**IHP**gss
mNa_v_1.3α	A2ASI5	agq**C**p**E**gyi	yie**Q**r**K**tik	gid**D**mfn	pda**IHP**gss
rNa_v_1.3α	P08104	agq**C**p**E**gyi	yie**Q**r**K**tik	gid**D**mfn	pda**IHP**gss
* eeNa_v_1.4α	P02719	agk**C**p**E**gyt	yiw**R**r**R**vik	gvd**D**ifn	pdv**ENP**gtd
* hNa_v_1.4α	P35499	agh**C**p**E**gye	yie**Q**r**R**vir	gid**D**mfn	pnl**ENP**gts
mNa_v_1.4α	Q9ER60	agh**C**p**E**gye	yie**Q**r**R**vir	gid**D**mfn	ptl**ENP**gtn
rNa_v_1.4α	P15390	agh**C**p**E**gye	yie**Q**r**R**vir	gid**D**mfn	ptl**ENP**gtn
hNa_v_1.5α	Q14524	agt**C**p**E**gyr	yle**E**r**K**tik	gid**D**mfn	ptl****PNS****ngs
mNa_v_1.5α	Q9JJV9	agt**C**p**E**gyr	yle**E**r**K**tik	gid**D**mfn	pnl****PNS****ngs
rNa_v_1.5α	P15389	agt**C**p**E**gyr	yle**E**r**K**tik	gid**D**mfn	pnl****PNS****ngs
hNa_v_1.6α	Q9UQD0	agq**C**p**E**gyq	yie**Q**r**K**tir	gid**D**mfn	ldk**EHP**gsg
mNa_v_1.6α	Q9WTU3	agq**C**p**E**gfq	yie**Q**r**K**tir	gid**D**mfn	ldk**EHP**gsg
rNa_v_1.6α	O88420	agq**C**p**E**gfq	yie**Q**r**K**tir	gid**D**mfn	ldk**EHP**gsg
hNa_v_1.7α	Q15858	sgq**C**p**E**gyt	yie**R**k**K**tik	gin**D**mfn	pkk**VHP**gss
mNa_v_1.7α	Q62205	sgq**C**p**E**gye	yie**K**k**K**tik	gin**D**mfn	pkk**VHP**gss
rNa_v_1.7α	O08562	sgq**C**p**E**gyi	yie**K**k**K**tik	gin**D**mfn	pkk**VHP**gss
hNa_v_1.8α	Q9Y5Y9	sgh**C**p**D**gyi	yld**Q**k**P**tvk	gid**D**mfn	pnl****PNS****ngt
mNa_v_1.8α	Q6QIY3	agh**C**p**N**dyv	yle**E**k**P**rvk	gid**D**mfn	pnr****PNS****ngs
rNa_v_1.8α	Q62968	agh**C**p**G**gyv	yle**E**k**P**rvk	gid**D**mfn	pnl****PNS****ngs
hNa_v_1.9α	Q9UI33	nsa**C**s**I**qye	hle**N**q**P**kiq	gid**D**ifn	rsk**ESC**nss
mNa_v_1.9α	Q9R053	rrs**C**p**D**gst	nlp**S**r**P**qve	gid**D**ifn	esk**ASC**nss
rNa_v_1.9α	O88457	srp**C**p**N**gst	nlp**S**r**P**qve	gid**D**ifn	eak**EHC**nss
**Subunit**	**UniProt** **code ^1^**	**7, 8, 3, 4**	**1,5**	**6**	**2**
* eeNa_v_β1	A0A1L3MZ94	sng**A**c**VE**vds**D**tea	sck**MR**gev	mgs**K**ntf	yfd**R**tlt
* hNa_v_β1	Q07699	acg**G**c**VE**vds**E**tea	sck**RR**set	ngs**R**gtk	hvy**R**llf
mNa_v_β1	P97952	awg**G**c**VE**vds**D**tea	sck**RR**set	ngs**R**gtk	hvy**R**llf
rNa_v_β1	Q00954	awg**G**c**VE**vds**E**tea	sck**RR**set	ngs**R**gtk	hvy**R**llf
* hNa_v_β3	Q9NY72	cfp**V**c**VE**vps**E**tea	scm**KR**eev	ngs**K**dlq	nvs**R**efe
mNa_v_β3	Q8BHK2	cfp**V**c**VE**vps**E**tea	scm**KR**eev	ngs**K**dlq	nvs**R**efe
rNa_v_β3	Q9JK00	cfp**V**c**VE**vps**E**tea	scm**KR**eev	ngs**K**dlq	nvs**R**efe
hNa_v_β2	O60939	grs**M**e**VT**vpa**T**lnv	fns**CY**tvn	sgn**P**sky	yim**N**ppd
mNa_v_β2	Q56A07	grs**M**e**VT**apt**T**lsv	fns**CY**tvn	sgn**P**sky	yit**N**ppd
rNa_v_β2	P54900	grs**M**e**VT**vpt**T**lsv	fns****CY****tvn	sgn**P**sky	yit**N**ppd
hNa_v_β4	Q8IWT1	sle**V**s**VG**kat**D**iya	fss****CF****gfe	vgs**T**kek	hvk**N**pke
mNa_v_β4	Q7M729	sle**V**s**VG**kat**T**iya	fss****CY****gfe	egs**T**kek	fvr**N**pke
rNa_v_β4	Q7M730	sle**V**s**VG**kat**T**iya	fss****CY****gfe	egs**T**kek	fvr**N**pke

^1^https://www.uniprot.org.

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
