# Peer review of "Chemometric Models of Differential Amino Acids at the Navα and Navβ Interface of Mammalian Sodium Channel Isoforms"

_molecules, 2020, doi:10.3390/molecules25153551_

Round 1

Reviewer 1 Report

"Chemometric models of differential amino acids at the Nav α and β interface of mammalian sodium channel isoforms" by Villa-Diaz and colab.

The authors determined PPI of Na_v-alpha with different Na_v-beta subunits. The manuscript reported determining different theoretical parameters on the subunits and ECL. However, the rationale behind the design of the theoretical study is not clearly articulated.

Although I have been doing computational studies and analysis data using chemometrics, the manuscript is difficult to follow.

Is there any validation of the computational results with experimental observations or properties determined experimentally?

How the authors confirmed that the theoretical results would be reasonable in reflecting the properties of Na_v?

I wonder whether the manuscript should submit to a more specialized journal on this area.  

Author Response

Reviewer 1 Response

Comments and Suggestions for Authors

"Chemometric models of differential amino acids at the Nav a and ß interface of mammalian sodium channel isoforms" by Villa-Diaz and colab.

The authors determined PPI of Na_v-alpha with different Na_v-beta subunits. The manuscript reported determining different theoretical parameters on the subunits and ECL. However, the rationale behind the design of the theoretical study is not clearly articulated.

Yes. we agree, we have modified the text and the motivation in the Introduction, as well as the results to “bring back home” for the readers as well as the essential findings are now highlighted in dark blue.

Although I have been doing computational studies and analysis data using chemometrics, the manuscript is difficult to follow.

Yes, we agree, and indeed we feel much beholden that this reviewer provides us the opportunity to improve thanks to a second careful reading of our own work. We admit at the final stage we had to rush to get all different things done, texting, imaging and missing modelling, citing and so forth. Now, we present a well-thought totally new “overhaul” version.

Is there any validation of the computational results with experimental observations or properties determined experimentally?

Yes, we agree, so we added a fully automated model of the interface, IF, that we found on the internet on an academic Web server. It is reported as the second final topic in the SM. The Web-based IF is exactly the same as ours, which was independently elaborated by chemometrics.

How the authors confirmed that the theoretical results would be reasonable in reflecting the properties of Na_v? I wonder whether the manuscript should submit to a more specialized journal on this area.

Yes, we understand this, in this revised version, the chemometric procedure is embedded in context of the literature findings. We added hints to qualify our findings expressis verbis as such throughout the entire MS in dark blue. This way literature evidence and our computed evidence is separated and “mise en relief” as the French say, underscored or marked.

Please see a concise report of our thoughts and care-taking during the elaboration of the revised version, in the coverletter.

Reviewer 2 Report

Review report for “Chemometric models of differential amino acids at the Navα and Navβ interface of mammalian sodium channel isoforms” by Villa-Diaz et al.

This paper is a report of a research that did not give the expected results; however, the theoretical background is solid, the research design appropriate and the results well described.

The authors, starting from the nine sodium channel amino acid sequences, analyzed protein-protein interactions among the Navα subunit that forms the channel pore and the Navβ subunits. The aim was to increase the knowledge about the roles of voltage-gated sodium channels. Actually, the authors revealed some possible interactions but were unable to explain the specific function of the involved subunits.

 The theoretical background of the paper is not new, but solid. The research design is appropriate and the results are described in such a detailed way that may confuse readers. The fact that the paper was translated into English explains both typographical errors and the oddity of some sentences. 

According to me there are only minor corrections to be performed mainly due to the incorrect translation into English, for instance PPI sometimes reported as IPP.

As a result, I recommend its publication with minor revisions.

Author Response

Reviewer 2 Response

Comments and Suggestions for Authors

Review report for “Chemometric models of differential amino acids at the Nava and Navß interface of mammalian sodium channel isoforms” by Villa-Diaz et al.

We appreciate the evaluation, and share the idea that with the wealth of information provided in the field of chemometrics, it is also necessary to present practical application, in our case, the field of membrane channel research.

This paper is a report of a research that did not give the expected results; however, the theoretical background is solid, the research design appropriate and the results well described.

We totally share the critism, we actually failed in this aspect in our original version, probably due to missing time and distanciation to find better descriptive formulations. We show a detailed list of what was changed in the revised version, below.

The authors, starting from the nine sodium channel amino acid sequences, analyzed protein-protein interactions among the Nava subunit that forms the channel pore and the Navß subunits. The aim was to increase the knowledge about the roles of voltage-gated sodium channels. Actually, the authors revealed some possible interactions but were unable to explain the specific function of the involved subunits.

Yes, we agree, we have modified the text accordingly, please see the blue colored parts in the revised version. We took the critical points and discussed them. Some parts were deleted and others added. See details below, please.

The theoretical background of the paper is not new, but solid. The research design is appropriate and the results are described in such a detailed way that may confuse readers. The fact that the paper was translated into English explains both typographical errors and the oddity of some sentences.

Yes, we agree, we were motivated to carry out established applications for the sake of reliability and documentation, see SM, please.

According to me there are only minor corrections to be performed mainly due to the incorrect translation into English, for instance PPI sometimes reported as IPP.

Again we thank this reviewer for tolerating our bad English and not having rejected the work. The English now is fluent, with a large vocabulary and established phrasing. We analysed semantics and semiotics, we created our own symbols. In the introduction of a technical term, we added definitions, e.g. the isoforms, the subunits, alfa or beta, the PPI and the IF. We analysed the levels of mechanistic complexity and now, clearly state, if we mean topological overall aspects, or the monomeric vs. polymeric essence of subunits, or the atomic level of those events which we described. This way we feel that we correspond to what the reviewer is observing here.

As a result, I recommend its publication with minor revisions.

Yes, we took the advice very serious and here is hope that we achieved the required total revision.

Round 2

Reviewer 1 Report

The authors tried very hard to address the previous comments. The manuscript is slightly easier to read, but still difficult to follow. Taking section 1.4 as example, the authors tried to articulate the contributions of the manuscript, which improved a lot from previous version, but it is still difficult to follow.

In the first paragraph (line 94 – 100), the authors frame the research question of this manuscript as describing mechanistic behavior of Na_v-alpha/Na_v-beta to shed light on the modulation by the nine isoforms….. Does that mean the authors tries to identify essential interactions between Na_v-alpha and Na_v-beta that give rise to ion selecting properties of the sodium channels? How does the calculations (structural determinations, amino acid properties analysis etc.)  provide the information need to address the above research question is not mentioned.

In the paragraph at line 101-104. It describe the topology of Na_v-alpha which composed of structures outside the cell, on the cell membrane and a part inside the cell. I would think this paragraph would be describing the present understanding of Na_v-alpha rather than the contribution from this work. Should this paragraph be put under section 1.2?

The sentence at line 107 “The term PPI implies that both pairs were always treated in parallel for all three Mammalian species to provide a total and systematic view on chemometric patterns.” is typical example that create more problems than it solves. Does it tries to define PPI? Or is this the conclusion of the present study? Why PPI would relate to geometry of the complex in different species? Why the both pairs treated in parallel would provide a systematic view on chmometric patterns?

The next sentence is “While the (3D-) structures of the former pair have been experimentally elucidated (by cryo-109 electron microscopy or crystallography), no (3D-) structural information exists for the latter pair.” If I understand correctly, former pair refers to Na_v-alpha and Na_v-beta1 complex which has experimental structure and the later pair refers to Na_v-alpha and Na_v-beta3 complex.

I sincerely think that the manuscript should be interesting, but it is difficult to follow in its present form. Other issues such as Table 1, what does that mean PPI is yes or no? They don’t interact at all if N is designated? It might be good to let the reader know where is the Na_v_beta  located in the introduction (say, in Fig 1). For those who are not familiar with protein structure of sodium ion channels, a clear introduction would be important to guide readers.

Author Response

Please, read our Rebuttal Letter, labeled here as author-cover letter in PDF format. 
